# The Long-Term Impact of Moral Education on College Students’ Psychological Well-Being: A Longitudinal Study Revealing Multidimensional Synergistic Mechanisms

**DOI:** 10.3390/bs15020217

**Published:** 2025-02-15

**Authors:** Xianghui Tian, Yanlin Tang

**Affiliations:** 1School of Marxism, Liaoning University, Chongshan Campus, Shenyang 110036, China; 4022310098@smail.lnu.edu.cn; 2School of International Economics and Politics, Liaoning University, Chongshan Campus, Shenyang 110031, China

**Keywords:** moral education, psychological well-being, emotion regulation, gender differences, longitudinal study

## Abstract

Moral education significantly impacts the psychological well-being of college students by cultivating moral awareness, social responsibility, and moral values. However, existing studies have mainly focused on its cognitive effects (e.g., value formation and moral judgment development), with less attention to its impact on the affective level (e.g., positive and negative emotions). Longitudinal studies on its long-term effects are also limited. To address this gap, the present study conducted two waves of a longitudinal survey involving 423 Chinese university students. The results showed that baseline moral education levels significantly predicted subsequent psychological well-being, including higher psychological prosperity, more positive emotions, and fewer negative emotions. Additionally, gender moderated the relationship between moral education and psychological well-being: females showed significantly higher levels of well-being than males after receiving moral education. This study provides deeper insights into how moral education affects the psychological well-being of college students and offers theoretical and practical recommendations for enhancing emotion regulation and implementing psychological well-being interventions.

## 1. Introduction

In recent years, psychological well-being issues have increasingly become a core concern in higher education worldwide. Surveys indicate that, in China, 65% of college students experience varying degrees of psychological distress, including anxiety, depression, and mood disorders ([44]; [47]). College years represent a critical stage for the development of psychological well-being, as students face multiple challenges, such as academic pressure, interpersonal relationships, and career planning. These challenges place greater demands on educational interventions ([9]). Enhancing the psychological well-being of college students through systematic educational interventions has thus become a key focus of research in both academic and educational practice.

Moral education, as a comprehensive educational practice, aims to develop moral awareness, social responsibility, and moral values ([29]; [35]). Research has shown that these factors can enhance psychological well-being by strengthening students’ psychological resilience, emotional regulation, and sense of social belonging ([22]). However, existing studies have focused primarily on the cognitive dimension of moral education, while its long-term impact on the affective dimension remains underexplored. Additionally, most studies have relied on cross-sectional designs, with a lack of longitudinal studies to examine the long-term effects of moral education and its underlying mechanisms. This study proposes a theoretical model (Figure 1) that hypothesizes that moral education enhances students’ psychological resources (e.g., psychological resilience, emotional regulation, and sense of social belonging) by fostering moral awareness, social responsibility, and moral values. These improvements, in turn, promote psychological well-being by increasing psychological prosperity, increasing positive emotions, and reducing negative emotions. Unlike previous studies with single-dimensional analyses, this model highlights the interactions between multidimensional factors and their combined effects on psychological well-being ([16]; [13]). In addition, this study incorporates gender as a moderating variable to explore the differential effects of moral education across gender groups. The proposed model not only deepens the theoretical understanding of the relationship between moral education and psychological well-being but provides a systematic reference for educational practices. Using a longitudinal design and a sample of Chinese college students, this study systematically investigates the long-term effects of moral education on psychological prosperity, positive emotions, and negative emotions, while analyzing the moderating role of gender.

Based on the literature and theoretical derivations, the following hypotheses are proposed:

**H1.** 
*Moral awareness significantly increases the sense of psychological prosperity and positive emotions while reducing negative emotions.*


**H2.** 
*Social responsibility significantly enhances psychological prosperity and positive emotions while reducing negative emotions.*


**H3.** 
*Moral values significantly promote feelings of psychological prosperity and positive emotions and have a positive effect on reducing negative emotions.*


**H4.** 
*Gender moderates the relationship between moral education and psychological well-being, with females being significantly more likely than males to experience mood improvement.*


**H5.** 
*There is a synergistic effect among moral awareness, social responsibility, and moral values, which collectively enhance psychological well-being indicators by improving students’ psychological resources.*


## 2. Materials and Methods

### 2.1. Literature Review

#### 2.1.1. Moral Awareness and Psychological Well-Being

Moral awareness refers to an individual’s understanding of moral norms and their willingness to consciously practice moral behavior. This awareness is particularly significant in the student population, as they are in a critical stage of value formation. The development of moral awareness not only enhances students’ sense of belonging and identity within society but promotes their psychological well-being ([40]; [17]). Studies have shown that students with greater moral awareness tend to exhibit greater psychological resilience and emotional stability. This is attributed to the positive psychological resources gained through moral practices, such as the satisfaction and inner balance derived from engaging in helpful behaviors and moral reflection. These experiences contribute to an enhanced sense of self-worth ([5]). Additionally, moral behavior helps students establish harmonious interpersonal relationships and reduces feelings of loneliness and conflict, effectively alleviating anxiety and depression ([36]). Furthermore, moral awareness fosters the formation of optimistic and positive emotions, making students more inclined to approach life challenges in a constructive and proactive manner. It also enables them to mitigate negative emotions caused by internal conflicts or social pressures through self-regulation and moral actions. This finding indicates that moral awareness not only improves students’ psychological well-being but provides essential psychological support, helping them maintain a positive mindset and behavior in complex social environments ([25]).

Research on moral education and psychological well-being has focused primarily on Western education systems, particularly in the United States and Europe. Much of this research emphasizes the relationship between the goals of moral education and psychological well-being in different cultural contexts ([6]). For instance, moral education in the United States often emphasizes individual responsibility, critical thinking, and the ability to reflect on and address social injustice, whereas European countries place a greater emphasis on fostering multicultural understanding and social inclusion ([49]). These studies suggest that, in individualistic cultures, moral education tends to promote psychological resilience and social adaptability ([39]; [30]). However, these characteristics may not fully apply to countries with predominantly collectivist cultures, such as China, where moral education emphasizes group interests, social responsibility, and the transmission of traditional values. Given the unique context of Chinese culture, which prioritizes group collaboration and social harmony, this study explores how moral education can enhance psychological well-being by improving students’ sense of psychological prosperity and emotional regulation, thus addressing gaps in the literature. Moreover, the multidimensional synergistic mechanism proposed in this study provides a theoretical foundation for moral education practices in diverse cultural contexts and offers a basis for future cross-cultural comparative research.

#### 2.1.2. Social Responsibility and Psychological Well-Being

A sense of social responsibility refers to an individual’s understanding of social obligations and their willingness to fulfill these obligations in a positive manner. It reflects an individual’s tendency to take responsibility, care for others, and actively participate in social welfare when addressing societal issues ([33]; [48]). Among college students, cultivating a sense of social responsibility can help them develop positive social cognition and behavioral patterns, which, in turn, positively affect their psychological well-being ([7]). Studies have shown that a stronger sense of social responsibility significantly enhances individual psychological well-being, particularly by improving emotional regulation and social adaptability. Students who engage in volunteer activities or community service experience a sense of self-worth through these practices, which not only increases their self-esteem but strengthens their sense of belonging to society and their connections with others. This, in turn, improves their overall psychological well-being ([2]). Furthermore, college students with a heightened sense of social responsibility are often more concerned with social justice and the well-being of others. This concern motivates them to take positive action to address social problems. As a result, they experience greater life satisfaction and a greater sense of well-being ([23]).

The influence of social responsibility on psychological well-being operates primarily on two levels. On the one hand, participating in social practice activities allows students to establish a broad social support system, providing emotional support and recognition. This support helps alleviate feelings of isolation and psychological pressure, thereby reducing the emergence of negative emotions ([21]). On the other hand, a sense of social responsibility motivates individuals to adopt more positive and effective coping strategies when facing adversity, reducing feelings of helplessness and alleviating anxiety ([31]). Research indicates that students with a stronger sense of social responsibility not only demonstrate higher levels of psychological prosperity but exhibit a more stable emotional state and stronger adaptability when managing academic and life stress. The emotion-regulating function of social responsibility enables individuals to find psychological comfort by enhancing social support and reinforcing their sense of self-worth during negative experiences. This, in turn, significantly mitigates the negative effects of negative emotions on psychological well-being ([26]).

#### 2.1.3. Moral Values and Psychological Well-Being

Moral values serve as the ethical foundation for individual behavioral decisions, reflecting an individual’s understanding and internalization of judgments about good and evil, social norms, and interpersonal fairness and justice ([11]). Among college students, higher moral values are often strongly associated with better psychological well-being outcomes ([45]; [42]). Research indicates that college students with stronger moral values tend to exhibit greater emotional resilience and internal stability when confronted with moral conflicts or social pressures. This resilience enables them to manage life uncertainties and challenges more effectively ([50]). For example, when faced with moral dilemmas, students can rely on their internal moral code to make rational decisions, which not only alleviates negative emotions, such as guilt and anxiety, but enhances their self-esteem and trust in others. Additionally, engaging in moral behaviors, such as honesty, fairness, and empathy, fosters positive interpersonal relationships. These healthy social interactions, in turn, have a further positive effect on psychological well-being ([3]).

The influence of moral values on psychological well-being is reflected in two key areas: emotional regulation and interpersonal relationships. First, individuals with strong moral values tend to exhibit greater maturity in managing their emotions. They are more likely to adopt a constructive attitude when facing stress or challenges rather than resorting to avoidance ([50]; [1]). Second, college students with high moral values demonstrate higher levels of empathy and cooperation in their interpersonal interactions. These positive interactions help reduce feelings of loneliness and alleviate social anxiety ([32]; [51]). By fostering healthy social relationships, individuals with strong moral values gain greater emotional support and social recognition, which further enhances their psychological well-being. Thus, moral values not only help individuals maintain emotional balance when navigating moral dilemmas but play a critical role in strengthening social resilience, emotional regulation, and their overall sense of psychological prosperity ([10]).

#### 2.1.4. The Role of Gender Differences in the Relationship Between Moral Education and Psychological Well-Being

Gender differences play a significant role in the relationship between moral education and psychological well-being, particularly in areas such as emotional regulation, social role perception, and the degree of internalization of moral values. Research has shown that women tend to focus more on an ethic of care in moral education, exhibiting higher levels of empathy and emotional sensitivity. As a result, they rely more on emotional support and the quality of social relationships for their psychological well-being ([19]). By contrast, men tend to be more oriented toward an ethic of rights and responsibilities and more likely to alleviate psychological stress through problem-solving strategies ([41]). These differences suggest that moral education should adopt gender-specific strategies to promote psychological well-being. For females, the emphasis could be on emotional regulation and enhancing their sense of social belonging; whereas, for males, the focus might be on developing a sense of responsibility and improving problem-solving skills ([20]). Additionally, gender differences may influence how individuals respond to moral conflicts and the long-term effects of these conflicts on their psychological well-being. This highlights the need for moral education to be tailored to individual needs, providing a theoretical foundation for its personalized implementation ([27]).

### 2.2. Methods

#### 2.2.1. Sample Selection

This study employed a longitudinal design to examine the long-term effects of moral education on the psychological well-being of college students in China, along with gender differences, through a two-wave survey. Students from four comprehensive schools, representing a wide range of fields—including science and engineering, social sciences, humanities, business, and education—were recruited through random sampling. The initial sample consisted of 501 students, with a final valid sample of 423 (59.8% female), aged 18–25 years. To ensure statistical power, the research team conducted power analyses via G*Power software 3.1.9.7 (2022), assuming a medium effect size (Cohen’s f^2^ = 0.15), a significance level of α = 0.05, and a statistical power of 1 − β = 0.95. The analysis revealed that the minimum sample size for linear regression analyses would be 138 individuals. Accounting for potential sample attrition, baseline recruitment aimed for no fewer than 300 participants. Ultimately, 501 individuals completed the first wave of questionnaires, and 423 participants provided complete data across two measurements separated by six months. Four attention-check items (e.g., “Option 1 = completely disagree”) were included in the questionnaire to identify inattentive respondents and ensure data quality. In this study, data on moral education experiences and psychological well-being were collected through two measurement waves, and the moderating role of gender in the relationship between the two was analyzed. Careful control of sample size and data quality provided a solid foundation for subsequent analyses.

#### 2.2.2. Research Instruments

The instruments used in this study included the moral education questionnaire (Appendix A), the Sense of Psychological Prosperity Scale (Flourishing Scale), and the Scale of Positive and Negative Emotions (SPANE). The questionnaire assesses the moral education experiences of college students, covering the core dimensions of moral awareness, social responsibility, and moral values ([15]). Psychological prosperity was measured via the Sense of Psychological Prosperity Scale developed by [12] ([12]). This scale consists of eight items addressing aspects of self-growth, life goals, and interpersonal relationships. The responses are based on a 7-point scale (1 = strongly disagree, 7 = strongly agree), with higher total scores indicating a greater sense of psychological prosperity ([43]). Positive and negative emotions were measured via the Scale of Positive and Negative Emotions (SPANE), which includes six positive and six negative emotion items. These items are scored on a 5-point scale (1 = never, 5 = always), with mean scores calculated separately for positive and negative emotions ([24]). The validity and reliability of these instruments have been confirmed in several countries, including Singapore, the United States, China, and Germany, demonstrating their applicability to psychological well-being research across different cultural contexts.

#### 2.2.3. Data Collection

This study was conducted in two phases, with data collected at baseline (T1) and six months later at follow-up (T2). The aim was to explore the long-term effects of moral education on the psychological well-being of Chinese college students and its mechanisms of action.

Phase 1: Baseline Data Collection (T1)

The research team first sent a recruitment notice to the target population through the student management systems of the four schools, inviting students to participate in this study. All participants were required to read and electronically sign an informed consent form, which outlined the voluntary nature of participation, anonymity, and data confidentiality. The questionnaires and scales were distributed via an online platform, and participants were instructed to complete them within a specified time frame. The baseline phase focused on collecting students’ basic demographic information, moral awareness, social responsibility, moral values, sense of psychological prosperity, and positive and negative emotions. After the data collection was completed, the research team cleaned the data by eliminating invalid questionnaires and tested the reliability and validity of the measurement instruments to ensure high data quality.

Phase 2: Follow-up Data Collection (T2)

After the baseline data collection was completed, the research team randomly assigned participants to an experimental group (211 students) or a control group (212 students). Students in the experimental group participated in a six-month moral education intervention during the follow-up period, which consisted of two 1.5-h sessions per week, with core modules including moral awareness development workshops, social responsibility enhancement courses, and participation in social practice activities (e.g., volunteering and community projects) (see Appendix B for details).The design of these modules is based on real-life problem cases provided by ChopMelon Net (Appendix C), which compiles typical problem scenarios from the educational and psychosocial fields. These cases cover topics such as campus environmental protection, intergenerational communication, and social welfare (e.g., “Should we prioritize helping friends and relatives or strangers?” Case No. KG2024018; “Moral Dilemma of Visiting Nursing Homes”, Case No. KG2024534). Through case discussions and practical activities based on real-life issues, students can better understand and internalize the core content of moral education.

By contrast, the control group did not receive any intervention and only participated in their regular learning activities. The six-month intervention period was chosen to observe the effects of the moral education intervention on students’ psychological well-being. This duration not only corresponds to one academic semester but provides sufficient time for the experimental group to engage in the intervention sessions and internalize the moral education content.

The follow-up phase (T2) took place at the end of the intervention period, when the research team distributed a second wave of questionnaires and scales to participants in both the experimental and control groups via the same online platform. The content of the follow-up assessment was consistent with that of the baseline phase, primarily measuring psychological well-being indicators, such as a sense of psychological prosperity, positive emotions, and negative emotions. To ensure data consistency before and after the intervention, the research team used unique codes to match data from the two waves. To minimize sample attrition, participants were sent reminders via text messages and emails during the follow-up period, and a flexible time window was provided for completing the questionnaires in both the experimental and control groups. After data collection was completed, all the data were processed and stored via unique codes to ensure privacy and data security. Personally identifiable information was removed prior to data analysis to maintain anonymity. Ultimately, psychological well-being data from the experimental and control groups will be used for comparative analyses to evaluate the effectiveness of the moral education intervention and to explore its long-term effects on psychological prosperity and emotional regulation.

#### 2.2.4. Data Analysis

This study examines the long-term effects of moral education on college students’ psychological well-being and how gender differences moderate the relationship between moral education and psychological well-being through a multi-step data analysis process. The data were first pre-processed, which included handling missing values, conducting reliability and validity tests, and matching the T1 and T2 data. Descriptive statistical analyses were then performed to calculate the means and standard deviations of the variables and to test the comparability of the experimental and control groups at the baseline stage. The effectiveness of the moral education intervention was assessed via paired-samples *t*-tests and linear regression analyses. Interaction terms were included in the regression models to explore the moderating effect of gender. Additionally, multivariate mediation analysis was conducted to examine the mediating roles of moral awareness, social responsibility, and moral values in the relationship between the intervention and psychological well-being. Finally, repeated-measures ANOVA and data visualization techniques were applied to demonstrate trends in psychological well-being indicators between the experimental and control groups, providing a clear depiction of the intervention’s effects.

## 3. Results

### 3.1. Descriptive Statistics and Baseline Comparison

A total of 423 college students participated in this study, including 211 students (60% female) in the experimental group and 212 students (58% female) in the control group. The participants ranged in age from 18 to 25 years, with a mean age of 21.3 years (standard deviation = 1.2). At the baseline stage, there were no significant differences between the experimental and control groups in terms of gender ratio or age distribution (*p* > 0.05).

As shown in Table 1 and Table 2, there were no significant differences between the experimental and control groups in terms of the moral education variables (moral awareness, sense of social responsibility, and moral values) or the psychological well-being variables (sense of psychological prosperity, positive emotions, and negative emotions), with all *p*-values > 0.05. These findings indicate that the two groups had similar levels of psychological well-being and moral education prior to the intervention, making them comparable.

### 3.2. Validation of Intervention Effects

After the intervention (T2), the psychological well-being indicators of the experimental and control groups exhibited different trends. The results of paired-samples *t*-tests revealed the following results.

As shown in Table 3, the experimental group demonstrated significant improvements in psychological well-being after the intervention. The sense of psychological prosperity increased from a mean value of 5.45 to 6.02, positive emotions increased from 4.28 to 4.75, and negative emotions decreased from 2.38 to 1.95. All the changes were statistically significant, indicating that the moral education intervention effectively enhanced the sense of psychological prosperity and positive emotions of the experimental group while significantly reducing their negative emotions. By contrast, changes in the psychological well-being indicators of the control group were minimal. The sense of psychological prosperity increased only slightly from 5.42 to 5.50, positive emotions increased from 4.22 to 4.30, and negative emotions decreased marginally from 2.40 to 2.35. None of these changes reached statistical significance. These findings suggest that the moral education intervention had a positive and significant effect on the psychological well-being of the experimental group, whereas the control group showed insignificant changes in the psychological well-being indicators without the intervention.

In addition, to further assess the practical significance of the intervention effect, the present study calculated the effect sizes of the psychological well-being indicators before and after the intervention in the experimental group (Cohen’s d) to more clearly quantify the actual impact of the moral education intervention. The formula for the calculation of Cohen’s d is as follows: $$d=\frac{M_{T2}-M_{T1}}{SD_{pooled}},$$
where $M_{T2}$ an $M_{T1}$ are the post-intervention and pre-intervention means, respectively, and $SD_{pooled}$ is based on the combined standard deviation calculated from the pre-intervention and post-intervention standard deviations, $$SD_{pooled}=\sqrt{\frac{SD_{T1}^{2}+SD_{T2}^{2}}{2}}$$

On the basis of the results of this formula, the effect size of psychological prosperity is 0.57. This result shows that, after six months of moral education intervention, the psychological prosperity of the experimental group significantly increased from the baseline stage, and the effect of the intervention reached a medium-to-large intensity. The effect size of positive emotions is 0.53, which also indicates that the intervention had a positive effect of medium intensity in enhancing the positive emotions of the students. Notably, the effect size for negative mood was −0.52, which demonstrated that the intervention significantly reduced negative mood among the members of the experimental group, with a medium strength effect.

As shown in Figure 2, the experimental group’s sense of psychological prosperity increased significantly after the intervention, from 5.45 at T1 to 6.02 at T2, demonstrating the significant effect of the moral education intervention in enhancing their sense of psychological prosperity. By contrast, the control group’s sense of psychological prosperity showed only a slight, non-significant increase from 5.42 at T1 to 5.50 at T2. These findings indicate that the intervention had a substantial positive effect on the experimental group, whereas the control group exhibited minimal change.

As shown in Figure 3, the positive emotions of the experimental group improved significantly after the intervention, increasing from 4.28 at T1 to 4.75 at T2. This reflects the effectiveness of the moral education intervention in enhancing positive emotions. By contrast, the control group’s positive emotions increased only slightly, from 4.22 at T1 to 4.30 at T2, a change that was not statistically significant. These results suggest that the intervention significantly improved positive emotions in the experimental group, whereas changes in the control group were minimal.

As shown in Figure 4, the negative emotions of the experimental group decreased significantly after the intervention, from 2.38 at T1 to 1.95 at T2, indicating that the intervention effectively alleviated negative emotions. By contrast, the control group’s negative emotions decreased only slightly, from 2.40 at T1 to 2.35 at T2, a non-significant change. These findings provide further evidence of the intervention’s success in reducing negative emotions in the experimental group, whereas the control group showed only subtle changes.

The results of the paired-samples *t*-test and repeated-measures ANOVA clearly demonstrate that the psychological well-being indicators—sense of psychological prosperity, positive emotions, and negative emotions—significantly improved in the experimental group after the intervention, whereas no significant changes were observed in the control group. These findings suggest that the six-month moral education intervention had a positive effect on the psychological well-being of college students.

### 3.3. Hypothesis Validation

**H1.** *Moral awareness significantly increases the sense of psychological prosperity and positive emotions while reducing negative emotions*.

Figure 5 shows that moral awareness scores significantly and positively predict feelings of psychological prosperity (β = 0.32, *p* < 0.001) and positive emotions (β = 0.28, *p* < 0.001), but significantly and negatively predict negative emotions (β = −0.29, *p* < 0.001). These results indicate that greater moral awareness is associated with greater psychological prosperity and positive emotions, as well as lower negative emotions. H1 was supported.

**H2.** 
*Social responsibility significantly enhances psychological prosperity and positive emotions while reducing negative emotions.*


Figure 6 shows that social responsibility significantly and positively predicts psychological prosperity (β = 0.40, *p* < 0.001) and positive emotions (β = 0.35, *p* < 0.001), but significantly and negatively predicts negative emotions (β = −0.33, *p* < 0.001). These findings suggest that higher levels of social responsibility are associated with improved psychological prosperity and positive emotions while effectively reducing negative emotions. H2 was supported.

**H3.** 
*Moral values significantly promote feelings of psychological prosperity and positive emotions and have a positive effect on reducing negative emotions.*


Figure 7 reveals that moral values significantly and positively predict psychological prosperity (β = 0.37, *p* < 0.001) and positive emotions (β = 0.31, *p* < 0.001), but significantly and negatively predict negative emotions (β = −0.28, *p* < 0.001). These results suggest that stronger moral values contribute to better psychological well-being outcomes, supporting H3.

**H4.** 
*Gender moderates the relationship between moral education and psychological well-being, with females being significantly more likely than males to experience mood improvement.*


Figure 8 shows a significant interaction effect of gender on the relationship between moral awareness and positive emotions (β = 0.18, *p* = 0.01), indicating that females experienced a significantly greater increase in positive emotions than males experienced after receiving moral education. Additionally, gender had a significant moderating effect on the relationship between moral education and negative emotions (β = −0.16, *p* = 0.02), with females experiencing a greater decrease in negative emotions. These findings suggest that gender plays a significant moderating role in the relationship between moral education and psychological well-being outcomes, partially supporting H4.

**H5.** 
*There is a synergistic effect among moral awareness, social responsibility, and moral values, which collectively enhance psychological well-being indicators by improving students’ psychological resources.*


Figure 9 illustrates that moral awareness, through the synergistic effects of social responsibility and moral values, had significant indirect effects on psychological well-being outcomes. Specifically, it had a positive effect on psychological prosperity (indirect effect = 0.15, *p* < 0.01) and positive emotions (indirect effect = 0.12, *p* < 0.01), as well as a negative effect on negative emotions (indirect effect = −0.13, *p* < 0.01). These findings confirm that the combined effects of moral awareness, social responsibility, and moral values contribute significantly to improvements in psychological well-being indicators, verifying H5.

In summary, through a series of linear regression and multivariate mediation analyses, this study verified all five hypotheses. Moral education, encompassing moral awareness, social responsibility, and moral values, was shown to have a significant positive effect on psychological well-being outcomes, including psychological prosperity, positive emotions, and negative emotions. Furthermore, gender, as a moderating variable, significantly influenced these relationships, with females demonstrating greater improvements in psychological well-being than males after the intervention. Finally, the three core dimensions of moral education exhibited synergistic effects that further enhanced psychological well-being outcomes. These findings provide strong support for the theoretical model proposed in this study and offer a theoretical basis for the practical application of moral education in improving psychological well-being.

## 4. Discussion

### 4.1. Effect of Moral Education on Psychological Well-Being

This study provides empirical evidence that moral education significantly improves college students’ psychological well-being by enhancing moral awareness, social responsibility, and moral values. The intervention was designed as a multidimensional approach, incorporating emotional regulation and social practice activities alongside moral awareness. Although these components are not traditionally considered part of moral education in many universities, they were included to address broader psychological well-being. The results show that moral education enhances psychological prosperity and positive emotions, while reducing negative emotions. These findings not only extend previous research on the cognitive benefits of moral education but highlight its positive impact on emotional regulation and social responsibility, which together contribute to overall psychological well-being. Future studies may explore the independent and combined effects of these variables to deepen our understanding of their roles in promoting psychological well-being.

Furthermore, these findings suggest that moral education not only strengthens emotional resilience at the individual level but has a broader impact on social cohesion by promoting positive behavioral habits ([46]). For example, by encouraging participation in voluntary activities and social welfare programs, moral education enables students to experience a greater sense of social support and psychological satisfaction in real-life contexts. Overall, these results underscore the integrative effect of moral education on college students’ psychological well-being and provide valuable insights for colleges and universities in designing effective interventions for psychological well-being interventions.

### 4.2. Potential Mechanisms and Contextual Factors

This study sheds light on the potential mechanisms through which moral education impacts psychological well-being. It was found that moral awareness, social responsibility, and moral values work synergistically to enhance college students’ psychological well-being ([34]). This effect likely operates through the strengthening of the individuals’ psychological resources, such as emotional regulation, psychological resilience, and social support. Additionally, the findings indicate that females experience more significant psychological improvements following moral education, particularly in terms of increased positive emotions and reduced negative emotions. This may be because females tend to place greater emphasis on interpersonal relationships and emotional regulation, key aspects of moral education. However, it is also worth considering whether these results suggest that the intervention should continue to emphasize gender-specific strategies, or whether future research could explore a more balanced approach, encouraging both males and females to equally develop emotional regulation, empathy, and responsibility. Such an approach could potentially reduce gender-based expectations and foster more well-rounded psychological development.

Cultural background also appears to play an important role in the relationship between moral education and psychological well-being. In the context of Chinese collectivist culture, moral education emphasizes group interests and social harmony, which may further amplify the role of social responsibility and a sense of belonging in promoting psychological well-being. Furthermore, traditional Chinese values—such as benevolence, love, and mutual assistance—may provide students with stronger ethical motivation, thereby contributing to improved psychological well-being outcomes ([8]). These contextual factors enrich our understanding of the mechanisms through which moral education operates and highlight its varying effects across different cultural contexts.

### 4.3. Limitations and Future Directions

While this study provides important findings, it has several limitations. First, the sample was limited to students from specific universities in China, which may restrict the generalizability of the results to other college-aged populations, especially in less selective universities. The universities included in this study may not fully represent the diverse socio-economic backgrounds or academic experiences of all students in China. This limitation should be taken into account, as it may influence the broader applicability of the findings. Future research could consider including a more diverse sample from various universities, both selective and non-selective, to better understand the generalizability of the effects of moral education on psychological well-being across different student populations. Second, although the questionnaire measures used in this study were validated for reliability, the subjectivity of self-reported psychological well-being indicators may introduce some bias. Additionally, it is worth noting that participants might be more inclined to provide socially desirable answers, which could influence the observed effects. Future studies could incorporate more objective measures, such as physiological indicators or third-party assessments, to reduce potential biases and provide a more comprehensive understanding of the intervention’s impact. Third, the intervention duration of six months was relatively short and did not allow for the examination of long-term effects. The intervention lasted for 72 h in total, with two sessions per week, each lasting 1.5 h. Future research could explore longer-term interventions to assess whether the effects of moral education are sustained over time. Finally, potential confounding variables, such as individual personality traits and family background, may have influenced the robustness of the results. These factors should be taken into account in future studies to better understand the underlying mechanisms at play.

Furthermore, it is important to consider that, even if the findings of this study hold in replications, attempts to implement these interventions more broadly might face challenges. These challenges could arise if the conditions under which this experiment was conducted do not hold on campuses in general, or if professors or educators do not believe that such interventions will work. As discussed by [18] ([18]) in *Spinning Wheels: The Politics of Urban School Reform*, practical implementation may fail if institutional or cultural conditions are not conducive to such interventions. Future research should address the potential barriers to implementation and explore how different institutional contexts, faculty beliefs, and local conditions might affect the success of moral education programs.

Future research could address these limitations in several ways. First, cross-cultural studies could be conducted to explore the effects of moral education on psychological well-being across diverse cultural contexts and to examine culturally moderating mechanisms ([37]). Second, long-term longitudinal studies are recommended to assess the sustained effects of moral education on psychological well-being and its influence on psychological development into adulthood ([28]). Additionally, future studies could incorporate physiological indicators (e.g., brain neural activity and hormone levels) alongside individual background variables to uncover the multilevel mechanisms through which moral education impacts psychological well-being ([14]).

### 4.4. Significance of Effect Sizes and Educational Practices

The effect size data indicate that moral education intervention not only theoretically enhances psychological well-being indicators but significantly contributes to the improvement of college students’ psychological well-being in a practical sense. In addition, these results provide an empirical basis for moral education intervention, demonstrating its strong applicability in enhancing students’ psychological resilience, emotion regulation ability, and overall well-being. By calculating effect sizes, this study further validates the practical value of the intervention and offers important theoretical references and practical guidance for integrating moral education with psychological well-being interventions in the future.

In terms of the significance of educational practice, the findings of this study offer practical suggestions for colleges and universities ([4]). For example, moral education could be combined with psychological well-being education to create more targeted course modules, such as emotional regulation workshops, ethical decision-making simulations, and social welfare practice programs. Furthermore, the results suggest that educational interventions should consider gender differences and provide personalized ethics education for male and female students ([38]). However, it is also important to reconsider whether emphasizing gender-specific interventions is the best approach. Future educational interventions could focus on creating a more balanced framework that encourages both male and female students to develop emotional regulation, empathy, responsibility, and logical decision-making skills. Such tailored strategies could maximize the positive effects of moral education while fostering a more inclusive and balanced approach to addressing students’ needs.

## 5. Conclusions

The findings of this study align with those of previous studies and further validate the significant contribution of moral education to psychological well-being. Specifically, moral awareness, social responsibility, and moral values were shown to significantly enhance college students’ sense of psychological prosperity and positive emotions through multidimensional pathways while effectively reducing negative emotions. These results reinforce the theoretical foundation of moral education as both a cognitive and an affective psychological well-being intervention. Moreover, by introducing mechanisms such as gender moderation and synergistic effects, this study provides new insights into the role of moral education in improving psychological well-being, offering an important contribution to academic research in this area.

On the basis of these findings, colleges and universities should place greater emphasis on the role of moral education in fostering students’ psychological well-being. Integrating moral education with psychological well-being education into curriculum design is highly recommended, with teaching modules focused on emotional regulation, social responsibility practices, and ethical decision-making. Additionally, exploring more diverse and personalized approaches to moral education is essential. Tailored intervention programs that address gender differences and individual characteristics can maximize the positive effects of moral education. Furthermore, the application of modern technology, such as virtual reality (VR) and artificial intelligence (AI), offers a promising avenue for enhancing moral education. By creating immersive and interactive experiences, schools can further amplify the psychological well-being benefits of moral education. This innovative approach, which combines traditional methods with modern technology, presents a new direction for the future of higher education.

## Figures and Tables

**Figure 1 behavsci-15-00217-f001:**
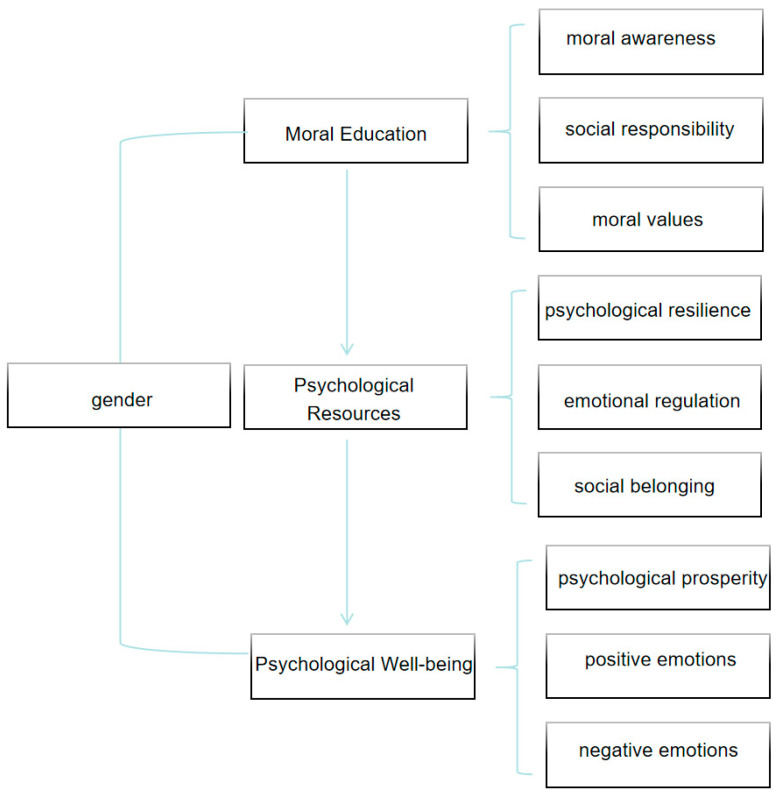
Theoretical model: Moral education and psychological well-being.

**Figure 2 behavsci-15-00217-f002:**
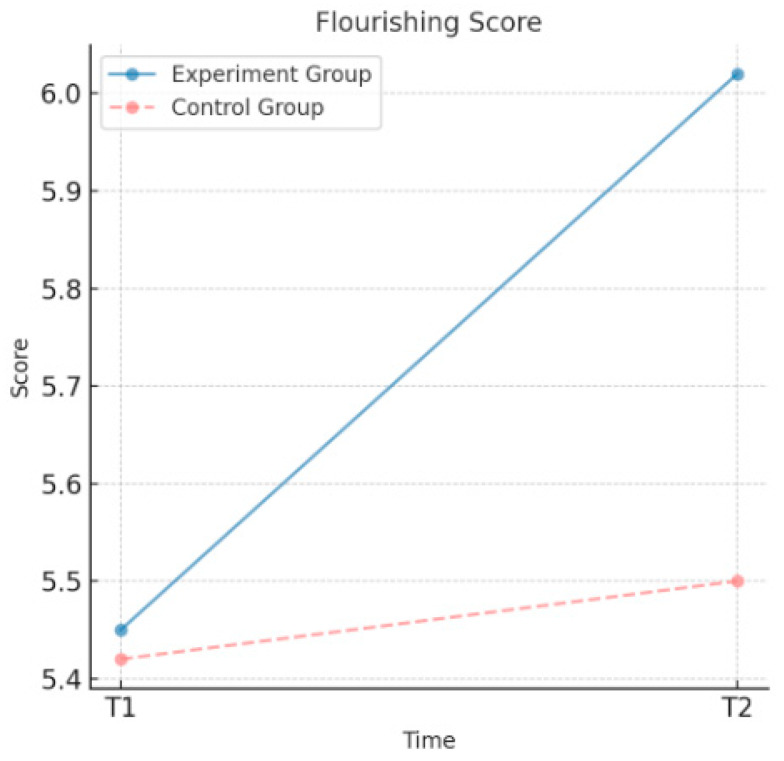
Changes in the sense of psychological prosperity over time.

**Figure 3 behavsci-15-00217-f003:**
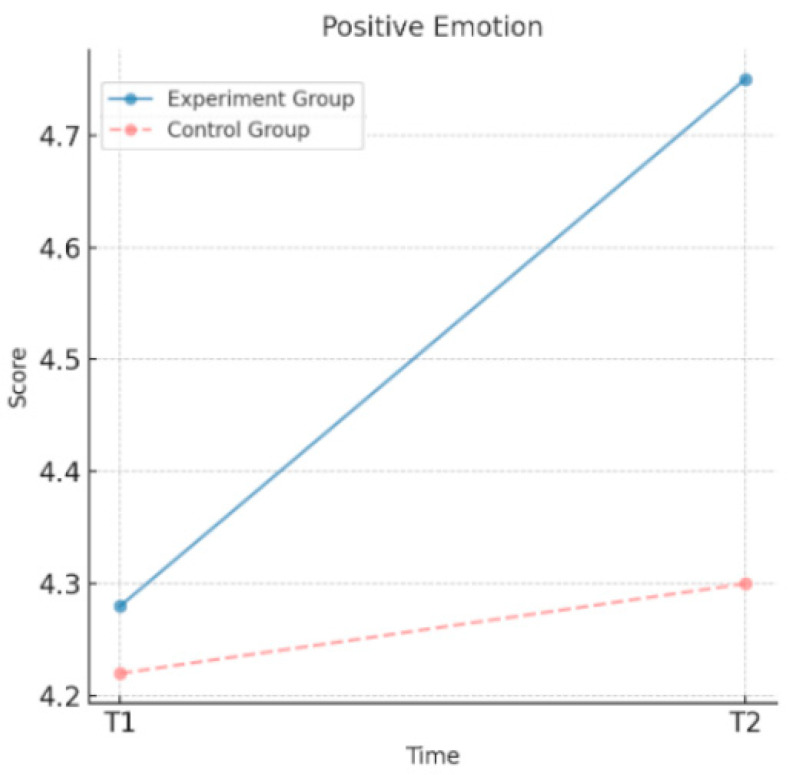
Changes in positive emotions over time.

**Figure 4 behavsci-15-00217-f004:**
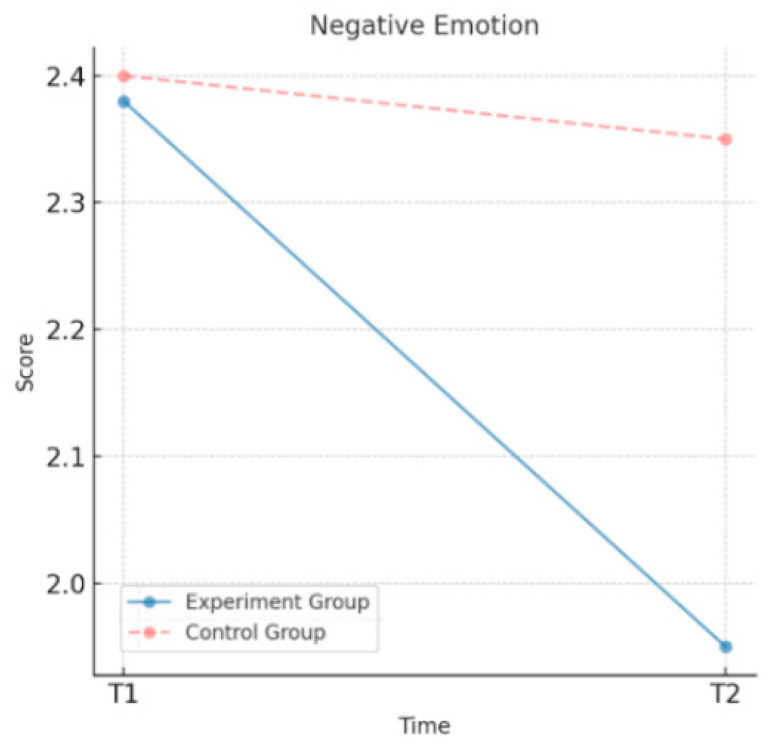
Changes in the negative emotions over time.

**Figure 5 behavsci-15-00217-f005:**
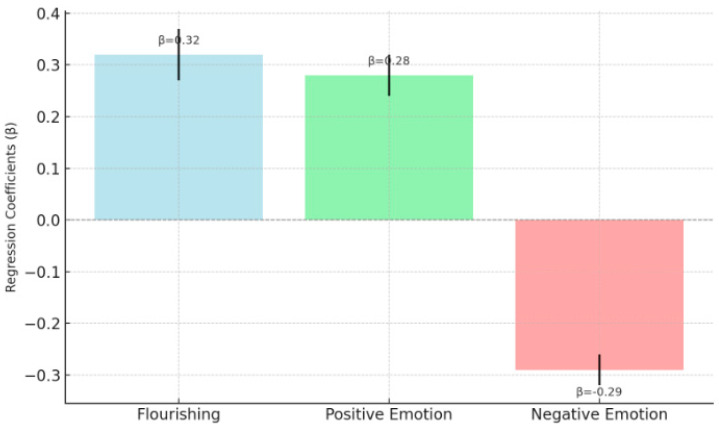
Effect of moral awareness on psychological well-being outcomes (H1).

**Figure 6 behavsci-15-00217-f006:**
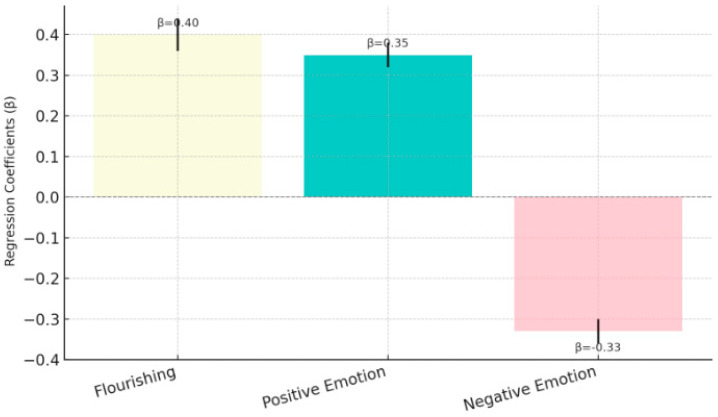
Effect of social responsibility on psychological well-being outcomes (H2).

**Figure 7 behavsci-15-00217-f007:**
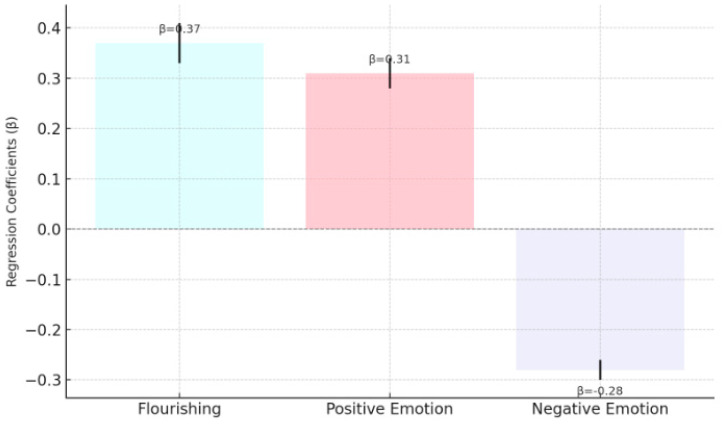
Effect of moral values on psychological well-being outcomes (H3).

**Figure 8 behavsci-15-00217-f008:**
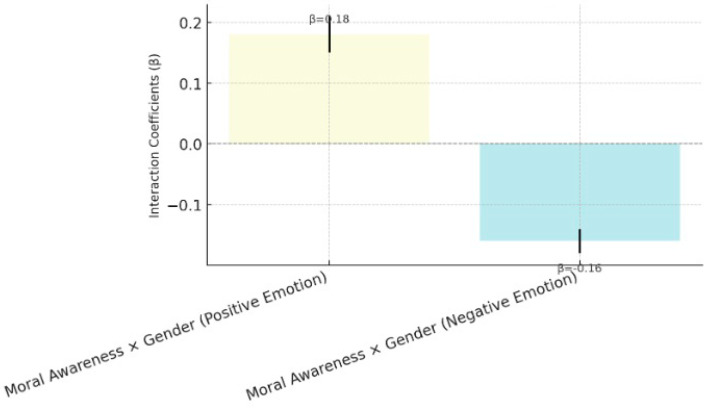
Moderating role of gender in psychological well-being outcomes (H4).

**Figure 9 behavsci-15-00217-f009:**
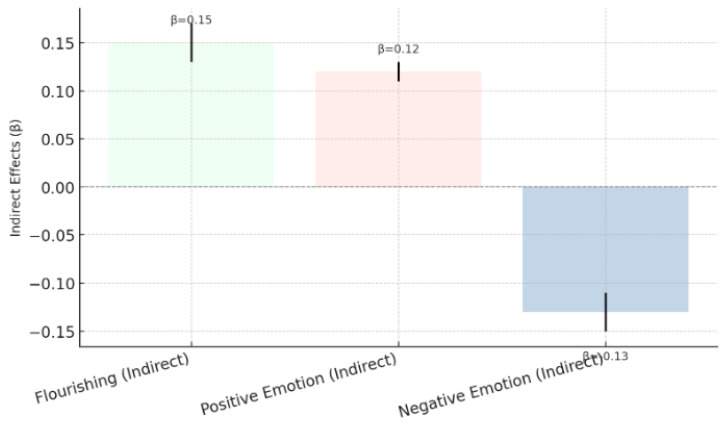
Synergistic effects of moral awareness, social responsibility, and moral values (H5).

**Table 1 behavsci-15-00217-t001:** Descriptive statistics of the experimental and control groups in terms of moral education indicators (T1).

Group/Indicators	Moral Awareness		Social Responsibility		Moral Values	
	M	SD	M	SD	M	SD
Experimental Groups	4.23	0.82	4.45	0.85	4.5	0.75
Control Groups	4.21	0.79	4.40	0.81	4.47	0.73
*p*-value	0.76		0.58		0.80	
Paired-sample *t*-test results	t = 0.23, *p* = 0.76		t = 0.21, *p* = 0.58		t = 0.13, *p* = 0.80	

**Table 2 behavsci-15-00217-t002:** Descriptive statistics of the experimental and control groups on psychological well-being indicators (T1).

Group/Indicators	Psychological Prosperity		Positive Emotions		Negative Emotions	
	M	SD	M	SD	M	SD
Experimental Groups	5.45	1.01	4.28	0.9	2.38	0.85
Control Groups	5.42	1.05	4.22	0.93	2.4	0.82
*p*-value	0.89		0.72		0.85	
Paired-sample *t*-test results	t = 0.13, *p* = 0.89		t = 0.14, *p* = 0.72		t = 0.11, *p* = 0.85	

**Table 3 behavsci-15-00217-t003:** Pre- and post-intervention statistics for the experimental and control groups.

Group/Times	Psychological Prosperity	Positive Emotions	Negative Emotions
Experimental Groups (T1)	5.45	4.28	2.38
Experimental Groups (T2)	6.02	4.75	1.95
Control Groups (T1)	5.42	4.22	2.4
Control Groups (T2)	5.5	4.3	2.35
*p*-value	0.03 *	0.04 *	0.01 *
Paired-sample *t*-test results	t = 2.12, *p* = 0.03	t = 2.25, *p* = 0.04	t = 3.13, *p* = 0.01

* *p* < 0.05.

## Data Availability

The data presented in this study are available on request from the corresponding author.

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
