# Peer review of "The Long-Term Impact of Moral Education on College Students’ Psychological Well-Being: A Longitudinal Study Revealing Multidimensional Synergistic Mechanisms"

_behavsci, 2025, doi:10.3390/bs15020217_

Round 1
Reviewer 1 Report
Comments and Suggestions for Authors
WHAT CAN BE IMPROVED.
The questionnaires used in the study should be included in an appendix.
The paper notes that students in the experimental group participated in a six-month moral education intervention but does not indicate how often each intervention occurred. More detail should be included—how often, how long etc.
COMMENTS
The intervention was also complex: 1. Moral Awareness Development Workshop, 2. Social responsibility enhancement course. 3. Social Practice Activities. 4. Emotional Regulation Enhancement Training. Though the paper suggests a synergistic relationship between these variables, future research ought to try and pull them apart. The second two variables, for example, are unlikely to be considered part of “moral education” in most universities, hence the claim that “moral education” has these long-term effects is highly ambiguous.
As well, the paper suggests that, with regard to moral education, “for females, the emphasis could be on emotional regulation and enhancing their sense of social belonging, whereas for males, the focus might be on developing a sense of responsibility and improving problem-solving skills.” However, it should be noted that even though the intervention indicated a more positive “emotional prosperity” in females, it is not clear that that is the most worthy goal if the result was a function of being validated in what one already believes. It is not clear, in other words, whether an education emphasizing the gender difference is optimal, rather than the reverse, i.e., educating so that males exhibit more care, while female exhibit more thought.
Author Response
|
Response to Reviewer 1 Comments |
|
Thank you very much for taking the time to review this manuscript,below are the detailed responses and the resubmitted manuscript. thank you again for your review. |
|
Comments 1: The questionnaires used in the study should be included in an appendix |
|
Response 1: Thank you for your valuable feedback. We have made the necessary revisions and added the full questionnaires to Appendix A of the manuscript to enhance clarity and completeness. Thank you again for your insightful suggestion.Page 18-20, lines 632-720 |
|
Comments 2: The paper notes that students in the experimental group participated in a six-month moral education intervention but does not indicate how often each intervention occurred. More detail should be included—how often, how long etc. |
|
Response 2: Thank you for your insightful suggestions. We have clarified the frequency and duration of the moral education intervention. In our study, students in the experimental group participated in a six-month moral education intervention, which consisted of two sessions per week, each lasting 1.5 hours. We have updated the manuscript to include these specific details in the methods section.Page 7, lines 266-271 |
|
Comments 3: The intervention was also complex: 1. Moral Awareness Development Workshop, 2. Social responsibility enhancement course. 3. Social Practice Activities. 4. Emotional Regulation Enhancement Training. Though the paper suggests a synergistic relationship between these variables, future research ought to try and pull them apart. The second two variables, for example, are unlikely to be considered part of “moral education” in most universities, hence the claim that “moral education” has these long-term effects is highly ambiguous. |
|
Response 3: We sincerely appreciate the reviewer’s thoughtful feedback. We understand your concerns about the complexity of the intervention and the relationships between the components of moral education, and we are grateful for your constructive suggestion. As you pointed out, emotional regulation enhancement training and social practice activities might not typically be considered part of "moral education" in many universities. We acknowledge this point, and in the revised manuscript, we have added this elaboration and clarification to the discussion section. While emotional regulation and social practice activities are not traditionally seen as parts of moral education, our design aimed to explore the multidimensional impact of moral education. These modules were intended to help students enhance emotional management and social responsibility, thereby promoting their psychological well-being. Regarding future research directions, we completely agree with your suggestion and plan to further examine these variables independently in future studies to provide a clearer understanding of the specific roles and long-term effects of moral education. Thank you again for your valuable insights, which will help us refine the research framework and theoretical assumptions further.Page 15, lines 474-486 |
|
Comments 4: As well, the paper suggests that, with regard to moral education, “for females, the emphasis could be on emotional regulation and enhancing their sense of social belonging, whereas for males, the focus might be on developing a sense of responsibility and improving problem-solving skills.” However, it should be noted that even though the intervention indicated a more positive “emotional prosperity” in females, it is not clear that that is the most worthy goal if the result was a function of being validated in what one already believes. It is not clear, in other words, whether an education emphasizing the gender difference is optimal, rather than the reverse, i.e., educating so that males exhibit more care, while female exhibit more thought. |
|
Response 4: Thank you for your insightful comment. We appreciate your observation regarding the gender-specific emphasis in the intervention, particularly in relation to emotional regulation for females and responsibility development for males. We agree that the observed improvements in emotional prosperity for females may not necessarily indicate the most appropriate or optimal goal, especially if the intervention simply reinforced existing beliefs or behaviors. We value your suggestion of exploring whether it might be more beneficial to approach the intervention with the aim of encouraging greater care in males and more analytical thinking in females, rather than reinforcing traditional gender roles. This is an important consideration, and we acknowledge that such an approach could offer more balanced and effective outcomes in promoting psychological well-being across both genders. We will reflect on this perspective in the discussion of our findings and consider it in the design of future interventions. Your feedback is invaluable, and we will explore this alternative approach in greater detail in future research.Page 15-16, lines 497-512;Page 17, lines 582-590 |
Reviewer 2 Report
Comments and Suggestions for Authors
The Long-Term impact of moral education on college students’ well-being: a longitudinal study revealing multidimensional synergistic mechanisms.
This is an interesting and seemingly---though I lack much knowledge in the area---original study which should be punished after revisions, none of which will be very difficult. It is not pathbreaking which is why I answered average for many of the evaluations, but is worth publishing, after revisions. In order of appearance.
First, there is a literature on how religion may improve well-being, in ways analogous to those suggested here. You need to at least cite Robert J Barro and Rachel M. McCleary’s The Wealth of Religions (Princeton, 2019) and make that point.
Second, were the subjects here representative of college-aged Chinese students as a whole? Discuss the representativeness of the sample. For example, are these highly selective universities? If not representative, how might that affect findings?
Third, regarding the intervention, how many hours did it take? More time would suggest greater commitment. In the limitations section, you should suggest that those going through the intervention might be more apt to provide socially desirable answers—that could be driving some of the statistical effects, and calls for more research.
Fourth, on the tables you should include T-test or Chi Square test results and p-values, to give us a feel for the impacts.
Fifth, you should offer the caveat, perhaps in the discussion, that even if these findings hold in replications, attempts to implement them might fail if conditions of this experiment do not hold on campuses generally, or if teachers (or in this case, professors) do not believe the interventions will work. There are many good citations you can use on this. One good one is Hess, Frederick M. (1997). Spinning Wheels: The politics of urban school reform (Washington, D.C., Brookings Institution).
Finally, go over the writing. There are some odd, at times redundant phrases such as, in the abstract “existing studies have primarily focused primarily…” I also wonder if it can be shortened a bit.
Generally, though, an interesting, clear study.
Author Response
|
Response to Reviewer 2 Comments |
|
Thank you very much for taking the time to review this manuscript,below are the detailed responses and the resubmitted manuscript. thank you again for your review. |
|
Comments 1: This is an interesting and seemingly---though I lack much knowledge in the area---original study which should be punished after revisions, none of which will be very difficult. It is not pathbreaking which is why I answered average for many of the evaluations, but is worth publishing, after revisions. In order of appearance. |
|
Response 1: Thank you very much for your constructive feedback. I am glad to hear that you find the study interesting and worth publishing after revisions. I appreciate your acknowledgment of the work’s originality, and I understand your assessment of its significance. I will carefully address the revisions you have suggested, and I am confident that these changes will improve the manuscript. |
|
Comments 2: First, there is a literature on how religion may improve well-being, in ways analogous to those suggested here. You need to at least cite Robert J Barro and Rachel M. McCleary’s The Wealth of Religions (Princeton, 2019) and make that point. |
|
Response 2: Thank you for your valuable suggestions. I appreciate your pointing out the relevant literature on how religion improves well-being. We have cited Robert J. Barro and Rachel M. McCleary's The Wealth of Religion (Princeton, 2019) in our revisions. Your suggestions will improve the depth of the manuscript, and I will make sure to address this in the revised edition.Page 22, lines 797 |
|
Comments 3: Second, were the subjects here representative of college-aged Chinese students as a whole? Discuss the representativeness of the sample. For example, are these highly selective universities? If not representative, how might that affect findings? |
|
Response 3: Thank you very much for your valuable feedback. The issue of the representativeness of the sample, as raised in your comment, is indeed an important point to address. In this study, the sample was drawn from several universities in China, which are influential but not necessarily representative of the entire population of college-aged students across the country. Therefore, the sample selection may have certain limitations, particularly in terms of student backgrounds, academic levels, and socio-economic status, all of which might affect their responses to the moral education intervention. To improve the representativeness of the sample, future studies could consider expanding the sample size and including universities from different regions, levels, and types of institutions to gain a more comprehensive understanding of the impact of moral education on the psychological well-being of college students. Thank you again for your feedback, we have further discussed the possible impact of sample representativeness on the results in our revisions, which are described in the discussion section.Page 16, lines 524-533 |
|
Comments 4: Third, regarding the intervention, how many hours did it take? More time would suggest greater commitment. In the limitations section, you should suggest that those going through the intervention might be more apt to provide socially desirable answers—that could be driving some of the statistical effects, and calls for more research. |
|
Response 4: Thank you for your valuable comment. Regarding the duration of the intervention, you raise an important point about how more time might suggest a greater commitment. The intervention in this study lasted for six months, with two sessions per week, each lasting 1.5 hours, totaling 72 hours. We will clarify this point in the revised manuscript and discuss how the duration of the intervention could potentially impact the findings. Additionally, your suggestion about the possibility of participants providing socially desirable answers is insightful. Given that participants may be more inclined to give responses that align with social expectations, this could indeed influence the statistical effects. In the limitations section, we will address this potential bias and suggest that future research could incorporate more objective measures, such as physiological data or third-party assessments, to further validate the effects of the intervention. Thank you again for your thoughtful feedback. We will revise the manuscript accordingly to strengthen the rigor of the study.Page 16, lines 533-542 |
|
Comments 5: Fourth, on the tables you should include T-test or Chi Square test results and p-values, to give us a feel for the impacts. |
|
Response 5: Thank you for your valuable feedback. As per your suggestion, we will include the results of paired-sample t-tests or Chi-square tests, along with p-values, in the tables to provide a clearer understanding of the impacts of the moral education intervention.Tables 1-3 on pages 8-9 |
|
Comments 6: Fifth, you should offer the caveat, perhaps in the discussion, that even if these findings hold in replications, attempts to implement them might fail if conditions of this experiment do not hold on campuses generally, or if teachers (or in this case, professors) do not believe the interventions will work. There are many good citations you can use on this. One good one is Hess, Frederick M. (1997). Spinning Wheels: The politics of urban school reform (Washington, D.C., Brookings Institution). |
|
Response 6: Thank you very much for your insightful comment. You raise an important point regarding the potential challenges and limitations of implementing these findings in real-world settings. Even if these findings hold in replications, the conditions in this experiment may not be easily replicated on campuses more generally, especially if professors or educators do not believe that the interventions will be effective. I fully agree with your perspective that such challenges need to be addressed in the discussion section.We will include this caveat in the discussion, noting the potential barriers to implementation and highlighting this limitation. We will also cite relevant literature, including the one you mentioned by Hess (1997), Spinning Wheels: The Politics of Urban School Reform, to support this point.Thank you again for your valuable feedback. We will incorporate this discussion in the revised version accordingly.Page 16-17, lines 548-557;Page 22, lines 831 |
|
Comments 7: Finally, go over the writing. There are some odd, at times redundant phrases such as, in the abstract “existing studies have primarily focused primarily…” I also wonder if it can be shortened a bit.Generally, though, an interesting, clear study. |
|
Response 7: Thank you for your careful review of the writing. I appreciate your point regarding redundant phrases, such as the repetition of "primarily" in the abstract ("existing studies have primarily focused primarily..."). I will revise this section to remove unnecessary repetition and make the expression more concise and fluent. Additionally, I will go through the entire manuscript to streamline the language, eliminating any redundant words or phrases, ensuring the writing is clear and precise.Your feedback is invaluable, and I will make sure to address these issues in the revision to improve the quality of the manuscript's language. |